# MUSE: The MUon Scattering Experiment

**E. Cline[1], J. Bernauer[1,2], E. J. Downie[3] and R. Gilman[4]⋆**

**1** Stony Brook University, Stony Brook, NY
**2** Riken BNL Research Center, Upton, NY
**3** The George Washington University, Washington, DC USA
**4** Rutgers, The State University of New Jersey, Piscataway, NJ

⋆ rgilman@physics.rutgers.edu

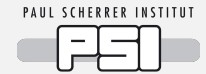
## Abstract

**MUSE is a high-precision muon scattering experiment aiming to determine the proton radius. Muon, electron, and pion scattering will be measured at the same time. Two-photon exchange corrections will be determined with data using both beam polarities.**

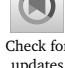

## 23.1 Introduction

The charge radius is a fundamental property of the proton. It is of interest to hadronic physicists as a test of calculations of proton structure. It is of interest to atomic physicists as it affects the determination of the Rydberg constant, and so is important in precision tests of quantum electrodynamics.

The charge radius can be determined using electromagnetic interactions in two ways. In atomic physics, the proton size changes the energies of S states by

$$\Delta E = \langle \Psi_S | \delta V | \Psi_S \rangle = \frac{2}{3} \pi \alpha \, |\Psi_S(0)|^2 \, r_p^2, \tag{23.1}$$

thus allowing the radius and Rydberg constant to be determined simultaneously by measuring pairs of transition energies. In electron-proton scattering, the differential cross section depends on the square of the form factor, which is the momentum-space charge distribution. The charge radius is extracted from the slope of the electric form factor $G_E$ at $Q^2 = 0$:

$$r_p^2 = -6 \frac{dG_E}{dQ^2} |_{Q^2=0}. \tag{23.2}$$

As the scattering data do not extend to $Q^2 = 0$, the radius is extracted from fits to measured cross sections.

In 2010 the proton charge radius was determined to be $0.84184 \pm 0.00067$ fm from a measurement of muonic hydrogen by the PSI CREMA collaboration [1]. This was quite puzzling as it was about $5\sigma$ smaller than the nearly order-of-magnitude less precise electronic measurements [2], which used both hydrogen spectroscopy and electron-proton scattering. This proton radius puzzle was quickly confirmed with reports from two new electron scattering measurements yielding $r_p = 0.879 \pm 0.008$ fm [3] and $0.875 \pm 0.010$ fm [4], and a second measurement of muonic hydrogen [5] that found $r_p = 0.84087 \pm 0.00039$ fm. New data are needed to resolve the proton radius puzzle, and a number of new experiments were developed [6–9]. Most aim to improve existing results, with new measurements of atomic hydrogen or electron-proton scattering. A new set of muonic atom measurements were also undertaken with other light nuclei.

## 23.2 The MUSE experiment

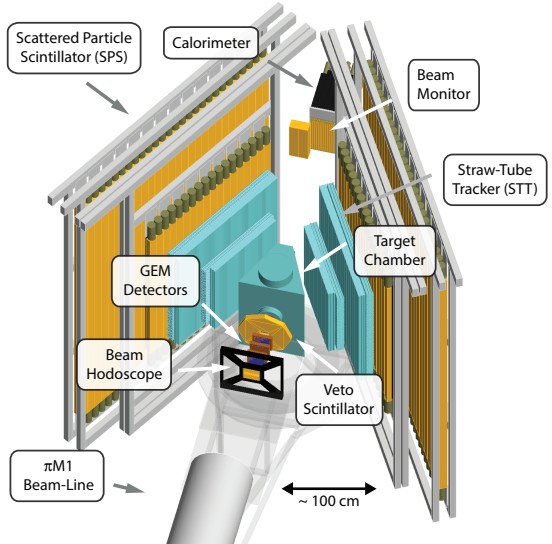

Figure 23.1: The MUSE experimental system. See text for details.

The MUon Scattering Experiment (MUSE) addresses the radius puzzle in a unique way. The intent is to extract the first precise proton radius measurement from muon-proton scattering. The experiment uses the PSI HIPA PiM1 channel [10, 11], which provides a secondary beam of pions, muons, and electrons. This enables simultaneous measurements of both electron and muon scattering, so that the extracted proton radii and the cross sections for the two reactions can be directly compared. The PiM1 channel can produce beams with similar beam properties for both polarities. A difference between the scattering probability for the two beam polarities would result from two-photon exchange, a higher-order correction to the interaction. This correction is expected to be small, O(0.1 – 1%), depending on kinematics, but it is difficult to calculate accurately. It might affect the determination of the radius.

Figure 23.1 shows the experimental apparatus, taken from the MUSE Geant4 simulation. Beam particles exiting the channel first pass through a beam hodoscope, which measures particle times. In conjunction with the accelerator RF signal, these times can be used to determine particle species. The beam next passes through GEM chambers, which measure the

beam-particle trajectories. A veto scintillator is used to suppress background events such as upstream beam particle decays in flight or scattering from the detectors, leading to particles passing through the vacuum chamber wall. The target system inside the vacuum chamber includes a liquid hydrogen cell, an empty cell, solid targets, and a beam focus monitor. The unscattered beam exits through a thin window, and reaches the downstream beam monitor and a calorimeter, which are used to study radiative corrections. Scattered particles exit through thin side windows, are tracked by the straw tube tracker, and their times measured with the scattered particle scintillators.

The PiM1 channel has been used previously for precise pion scattering measurements. This is feasible as pions are often the dominant species in the beam, and hadronic scattering cross sections can be orders-of-magnitude larger than electromagnetic cross sections. A primary challenge of MUSE is to measure precise cross sections for the smaller muonic component of the beam. The first aspect of the challenge is that previous determinations of beam properties concentrated on the pionic component of the beam, so the properties of the muonic and electronic components are not as well known. The second aspect is that the experimental system has to largely prescale away pion scattering to be able to efficiently measuring muon and electron scattering.

To address the challenge of beam properties, MUSE has undertaken a program of simulations and measurements. The first step is to simulate the particle production mechanisms at the M target. Charged pions are produced at the M target through $pC \rightarrow \pi^{\pm}X$ reactions. From the perspective of the PiM1 channel, the proton beam crosses the M target generating pions with an effective millimeter-sized source. Muons are produced by the decays in flight of those pions. Simulations show that the majority of the muons that will pass through the PiM1 channel are generated by pions that decay in the first few centimeters of flight, at an angle of nearly 90° in the pion rest frame. The effective muon source size is larger than the pion source size, but still only a few millimeters. Electrons and positrons are produced mainly by a sequence of reactions, with $pC \rightarrow \pi^0 X$ producing neutral pions, followed by the decay $\pi^0 \rightarrow \gamma\gamma$, and subsequently pair production in the M target via $\gamma C \rightarrow e^{\pm}X$. Geant4 simulations show that higher momentum electrons and positrons are only produced when all these processes are in the direction of the PiM1 channel. As a result, the effective source size remains very close to that for pions.

The source simulations generate charged particles that are input to the TURTLE [12] and G4 beamline [13] magnetic transport codes. These codes include the channel quadrupoles and dipoles, as well as apertures from beam pipes and jaws. The simulation describes well several measured properties of the beam, including the beam distributions in position and angle at the channel intermediate focal plane and at the scattering target position, and the variation of particle times at the scattering target with respect to accelerator RF as a function of momentum: the pion time distribution is wider than that for electrons or muons due to the interplay of faster speed vs longer flight path for higher-momentum particles within the channel. While the measured time distributions of all particles are quite similar, the muon distribution is predicted to be somewhat larger than the pion and electron distributions, indicating that extreme rays are more constrained in reality than in the simulation.

In addition to the particle trajectories, it is important to know the beam momentum at the 0.2% (0.3%) level for muons (electrons). The channel momentum resolution is better than this. The absolute momentum of the beam selected by the PiM1 channel is determined in 3 ways. First, dedicated time-of-flight measurements with changes of the beam hodoscope and beam monitor positions determine the pion and muon momenta to the 0.2 – 0.3% level. Second, the timing of particles in the beam hodoscope relative to the accelerator RF provides an independent momentum measurement at the same level.[1] Third, the dispersion of the

---

[1]This timing measurement also checks the beam momentum stability at the $\approx 0.1\% - 0.2\%$ level.

channel at the intermediate focal point, of 7 cm/%, combined with the dispersion of the beam from the intermediate focus to the scattering target of $\approx 9.5$ cm/%, provides a check of any momentum difference between the different particle species at the $\approx 0.1\%$ level, through the similarity of the measured beam spot positions.

The challenge of suppressing pion scattering while efficiently measuring muon and electron scattering is addressed by the MUSE trigger system. A first-level trigger FPGA identifies all particle species in the 3.5-MHz beam using the time difference between the beam-hodoscope signal and the accelerator RF signal. Other first-level triggers identify scattered particles and hits in the veto detector. The combination of these first-level triggers allows muon and electron scattering to be read out efficiently while suppressing pion scattering.

One important feature of MUSE will be the implementation of a blinded analysis in the cross section measurement. A Monte Carlo simulation is needed to determine precise cross sections, and from them the proton radius. The blinding will be accomplished primarily through modifying the simulation-derived weight factor, while encrypting the actual weights. Additionally, some small fraction of the tracks for different particle species will be thrown away as a function of angle, to prevent accidental unblinding by direct comparison of charge and / or particle species. This will be programmed to be reversed by the application of two encryption keys. Once the analysis is complete, the actual weights can be extracted and the physics analysis rerun.

A more detailed description of the MUSE system is available in [14]. Detailed publications are also available for the target [15] and the SiPM detectors [16].

## 23.3 Anticipated results

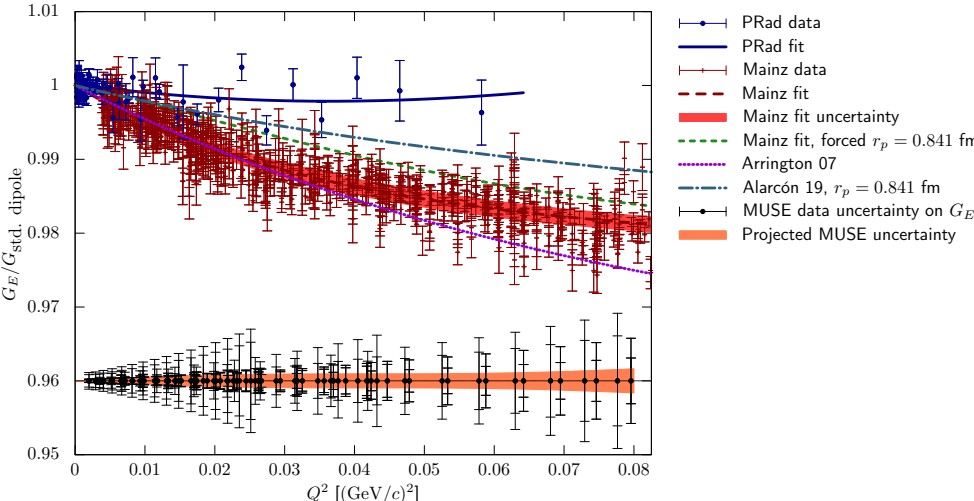

Figure 23.2: Anticipated data for $G_E$ from MUSE, arbitrarily placed at 0.96, compared to recent electron scattering experiments, and fits to these data, and to two world data fits. The MUSE data include both electron and muon points. The doubled uncertainty bars represent the uncertainties for + (inner bar) and - (outer bar) beam polarity. The muon and electron points are slightly offset due to the mass difference of muons and electrons. See text for further details.

With the planned 12 months of beam time, $4 \times 10^7$ $\mu^+$ ($2 \times 10^7$ $\mu^-$) scattering events are expected for MUSE. This should give better than 1% statistical precision for the cross section in almost all of the 16 planned angle bins at each of 3 beam momenta and two beam polarities.

Figure 23.2 shows the expected uncertainties for the determination of the electric form factor, $G_E$, from MUSE, together with the results from Mainz [3] and from PRad [17], along with two selected fits [18, 19]. The Arrington07 fit [18] is to older world data that are not shown, and has a large radius. The Alarcon19 curve [19] is a dispersively improved effective field theory calculation which has one free parameter, the radius, which can be fit, but here is chosen to be the muonic spectroscopy value. The green dashed "Mainz-fit" line is a fit to the Mainz data, but with the radius term set to the muonic spectroscopy value.

The experiments each measure in different kinematic regions, with MUSE at the lowest beam momentum and largest angles, and PRad at the highest beam momentum and smallest angles. The experiments also use different techniques. The more recent PRad measurement used a forward angle calorimeter to measure cross sections for 1.1 and 2.2 GeV beam energies at angles up to $\approx 7.5°$. The earlier Mainz measurements used magnetic spectrometers at larger scattering angles, with beam energies from 180 – 855 MeV. The Mainz and PRad data can be seen to diverge from each other, which probably indicates problems either with the experiments or with the radiative corrections. While the Mainz data are in good agreement with the Arrington fit to earlier data, neither the PRad nor the Mainz data agree with the prediction by Alarcon using the muonic radius. The expected MUSE uncertainties are competitive with those of the existing experiments. Muon scattering has much smaller single-photon radiative corrections, due to the larger muon mass, so any differences between muons and electrons might point to issues of radiative corrections or new physics.

The comparison of the cross sections for $+$ and $-$ polarities will yield a measurement of the two-photon exchange contribution, expected to be of similar size to the experimental uncertainties shown in Figure 23.2. The proton radius should be determined with an uncertainty of 0.006 – 0.010 fm, based on a sample of fits. The electron scattering data will have superior statistical precision, but larger systematic uncertainties due to radiative corrections. This should result in slightly better measurements for both the radius and the two-photon exchange contribution.

In addition to the electromagnetic scattering, pion cross sections need to be measured during MUSE to sufficiently characterize experimental backgrounds. The pion cross sections are interesting by themselves as a test of the application of chiral perturbation theory, to improve the existing $\pi N$ scattering database, and as a constraint on occasional speculations about undiscovered resonances in the $\pi N$ system. Because MUSE operates with a mixed beam, pion scattering will be measured in all MUSE kinematics at the same time as the electron and muon scattering. The experimental trigger includes beam particle information, which allows the pion scattering events to be pre-scaled to become a small fraction of the data set, while still recording on the order of $10^7$ events.

## 23.4 Outlook

A test of the full MUSE system in December 2019 led to several planned upgrades to make the system more robust. Due to the ongoing international public health crisis and its resulting impact on international travel, we were only able to partially complete the upgrades during 2020. We plan to complete the upgrades and start MUSE production data taking in 2021. With 12 months of data taking and analysis to be performed, we anticipate publication of first results in 2023/24. MUSE will be the first experiment to measure elastic muon-proton scattering in an appropriate kinematic region, with a precision sufficient to address the proton radius puzzle. The corresponding results for the simultaneously-measured electron scattering, will put a strong constraint on potential systematic uncertainties, and may help settle the discrepancies between the Mainz and PRad results. MUSE will be the only experiment that can directly measure with its own data the difference between electron and muon extractions of the radius, making it highly compelling.

## Acknowledgement

This work was supported in part by the U.S. National Science Foundation, grants PHY-1913653, 2012940 and 2012114.

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
