# Peer review of "MUSE: The MUon Scattering Experiment"

_SciPost Physics Proceedings, doi:SciPost Phys. Proc. 5, 023 (2021)_

## Round 1 · Referee Report · Adrian Signer (Referee 1) · 2021-4-9

Report

We (the editors Cy Hoffman, Klaus Kirch, Adrian Signer) had the
opportunity to review an earlier draft of the article and were in
communication with the authors before the submission. All our comments
and suggestions have been taken into account. Hence, we think the
paper can now be published in the current form.

---

## Round 1 · Referee Report · Claude Petitjean (Referee 2) · 2021-4-10

Strengths

1) good introduction + short review of r_p² measurements 2) comprehensive description of the experiment 3) good presentation of anticipated error bars

Weaknesses

1) Fig23.2 caption: no comments to the PRad&Mainz data and the theoretical curves 2) Anticipated results: unsufficient discussion of "PRad" vs. "Mainz" data and the relevance of the theoretical curves to the exp. data 3) pions: no program is given what needs to be measured to keep its background and systematics under control

Report

The authors present a comprehensive description of the MUSE experiment - presently in commissioning phase - and its goals toward elucidating the "proton radius puzzle". The paper is well written and referenced .Thus it fulfills all criteria for acceptance in the journal.

The chapter "Anticipated results" shows a very interesting figure of G_E/G_E_std.dipole vs. Q², but the data (PRad,Mainz) and the theoretical fit curves need to be better explained and discussed with respect to each other.
The pions - main component of the PiM1 beam - are treated only marginally although its role as backgrounds and origin of systematics may be quite significant. It should be emphasized in more detail what measurements and studies will be done.

Requested changes

1) Fig. 23.2 caption: explanation of exp. data and theory fits 2) Chapter "Anticipated results": The exp. data (PRad,Mainz) and theory fits should be explained and discussed more comprehensively. E.g. what is the "PRad" experiment and where does it stand with respect to the "Mainz" data? How helpful (relevant) are are the theoretical fits? 3) Pions (lines 145-149): present an exp. program of measurements and studies to be done.

Attachment

---

## Round 2 · Referee Report · Claude Petitjean · 2021-5-17

Report
The manuscript has been resubmitted with all the Improvements made as recommended by the referee. It is ready for publication.
Small comment: A typo in the first word of Figure 23.2 caption should be corrected.
Anonymous on 2021-05-26 [id 1468]
We thank the referee and editors for their work on reviewing this manuscript.

---

## Editorial Decision

published